# Sparse Black-Box Multimodal Attack
# for Vision-Language Adversary Generation

**Zhen Yu**[1][*], **Zhou Qin**[2][*], **Zhenhua Chen**[1][*],
**Meihui Lian**[2], **Haojun Fu**[2], **Weigao Wen**[2], **Hui Xue**[2], **Kun He**[1][†]

[1] School of Computer Science and Technology,
Huazhong University of Science and Technology, Wuhan, China
[2] Alibaba Group
baiding15@hust.edu.cn, brooklet60@hust.edu.cn

## Abstract

Deep neural networks have been widely applied in real-world scenarios, such as product restrictions on e-commerce and hate speech monitoring on social media, to ensure secure governance of various platforms. However, illegal merchants often deceive the detection models by adding large-scale perturbations to prohibited products, so as to earn illegal profits. Current adversarial attacks using imperceptible perturbations encounter challenges in simulating such adversarial behavior and evaluating the vulnerabilities of detection models to such perturbations. To address this issue, we propose a novel black-box multimodal attack, termed **Sparse M**ultimodal **At**tack (**SparseMA**), which leverages sparse perturbations to simulate the adversarial behavior exhibited by illegal merchants in the black-box scenario. Moreover, SparseMA bridges the gap between images and texts by treating the separated image patches and text words uniformly in the discrete space. Extensive experiments demonstrate that SparseMA can identify the vulnerability of the model to different modalities, outperforming existing multimodal attacks and unimodal attacks. SparseMA, which is the first proposed method for black-box multimodal attacks to our knowledge, would be used as an effective tool for evaluating the robustness of multimodal models to different modalities. Code is available at https://github.com/JHL-HUST/SparseMA.

## 1 Introduction

With the rapid development of Deep Neural Networks (DNNs), vision-language multimodal classification has been applied in various real-world applications, such as product restrictions on e-commerce (Zahavy et al., 2018) and hate speech monitoring on social media (Kiela et al., 2020), to ensure secure governance of various platforms.

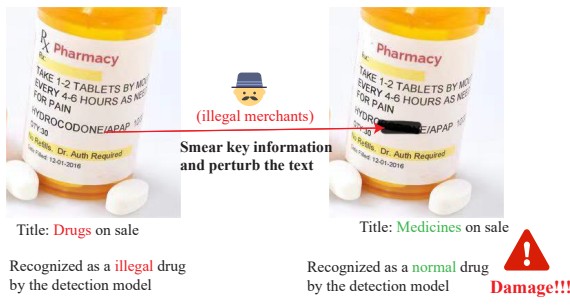

Figure 1: An example of a prohibited drug, where some key information is smeared by the illegal merchant, which is identified as compliant by the detection model.

However, there are always some illegal merchants who attempt to sell prohibited goods to make illegal profits, such as illegal drugs or smuggled goods, which violate the rules and terms of platforms. Due to the lack of professional knowledge in adversarial attacks (Szegedy et al., 2014; Papernot et al., 2016b), they often adopt large-scale and sparse perturbations, including smearing and mosaicing, to deceive detection models deployed by various companies, thereby posing a significant threat to the security of platforms, as shown in Figure 1.

Current adversarial attacks (Goodfellow et al., 2015; Papernot et al., 2016b) typically mislead the victim model by adding imperceptible perturbations to benign samples, which can not accurately evaluate the vulnerability of detection models to the adversarial behavior exhibited by illegal merchants. In addition, prior research has mainly focused on unimodal adversarial attacks in Computer Vision (CV) (Goodfellow et al., 2015; Madry et al., 2018) and Natural Language Processing (NLP) (Papernot et al., 2016b; Liang et al., 2018), with little attention paid to the vulnerability of multimodal models (Yu et al., 2020; Yang et al., 2021) that are more challenging but widely used in the real world. This motivates us to develop a multimodal adversarial attack that simulates physical adversarial behaviors

---

[*]The first three authors contributed equally.
[†]Corresponding author.

in the black-box scenario, which can help ensure secure governance of various platforms.

Existing multimodal attacks (Evtimov et al., 2020; Zhang et al., 2022) typically process images and texts independently due to the vastly different data attributes of the two modalities, such as continuous images *vs.* discrete texts. When employing different attack strategies to perturb images and texts, these methods are unable to effectively combine features from different input modalities, leading to relatively low attack performance. And these attacks run in white-box attack settings, which are almost impossible to apply to the real-world.

To address the above issue and simulate the sparse perturbations commonly used by illegal merchants, we propose to map them into the same discrete space to bridge the gap between continuous images and discrete texts. And then we perturb both images and texts simultaneously using the same sparse attack strategy. Although these sparse perturbations are indeed visible, they generally do not alter the semantics.

In this work, we propose a novel black-box multimodal attack, named **Sparse Multimodal Attack** (**SparseMA**). SparseMA splits the input image into multiple patches, then evaluates the impact of each patch in the image and each word in the input text on the victim model based on the output logits. After sorting all patches and words according to their impacts, we sequentially replace the original data with suitable candidates until an adversarial example is found. Note that SparseMA only needs access to the model output, making it more feasible in the real-world than white-box attacks, which require full access to the model, including architecture, parameter, gradient, output, *etc*.

To validate the effectiveness of the proposed SparseMA, we do comparison with one image attack, two text attacks, and two multimodal attacks on the task of vision-language multimodal classification. Extensive experiments demonstrate that SparseMA could identify the vulnerability of the model to different modalities, achieving a higher attack success rate than almost all the baselines. Moreover, SparseMA could generate better sparse adversarial examples with fewer perturbations, making them more similar to benign samples and more applicable to real-world scenarios. Through analyzing the impact of perturbations on the model's output in each modality, SparseMA can reveal the relative importance of each modal-ity in the model's decision-making process. This information is valuable for researchers to better understand the multimodal model's behavior and enhance its robustness against adversarial attacks.

## 2 Related Work

This section provides a brief overview of unimodal adversarial attacks on images or texts, as well as multimodal adversarial attacks.

### 2.1 Unimodal Adversarial Attack

**Image adversarial attacks**   Szegedy et al. (2014) first show the existence of adversarial examples. Subsequently, numerous works based on the $l_\infty$ or $l_2$ norm, including FGSM (Goodfellow et al., 2015), PGD (Madry et al., 2018), MIM (Dong et al., 2018), NIM (Lin et al., 2020), *etc*., are proposed to enhance the attack performance. JSMA (Papernot et al., 2016a) is the first to generate sparse perturbations by minimizing the $l_0$ norm. Croce and Hein (2019) introduce the $l_0$ norm variant of PGD, known as $PGD_0$, and a black-box sparse attack of CornerSearch that evaluates the saliency of each pixel based on the changes on logits. LocSearchAdv (Narodytska and Kasiviswanathan, 2017), ADMM (Zhao et al., 2019), and Sparse-RS (Croce et al., 2022) adopt local search, Bayes optimization, and random search algorithms, respectively, to search for the optimal sparse adversarial example in the black-box setting.

Despite the prosperity of sparse black-box attacks, they generally search for the optimal solution or evaluate the importance for each pixel, which require more than $10^6$ queries on images of size (224, 224). Even if ADMM and Sparse-RS optimize the query efficiency, tens of thousands of queries are still required, making it challenging to scale to large-scale datasets, such as ImageNet (Deng et al., 2009). In this work, SparseMA splits the image into multiple patches, and does not need to process each pixel. It allows us to achieve good attack performance with only a few hundred queries.

**Text adversarial attacks**   Existing text adversarial attacks typically modify the character, word, or sentence of the benign text to maximize the loss on the ground-truth class, among which word-level attacks (Li et al., 2020; Garg and Ramakrishnan, 2020; Li et al., 2021a; Maheshwary et al., 2021; Yu et al., 2022) show excellent performance. PWWS (Ren et al., 2019) and TextFooler (Jin et al., 2020) greedily substitute important words with syn-

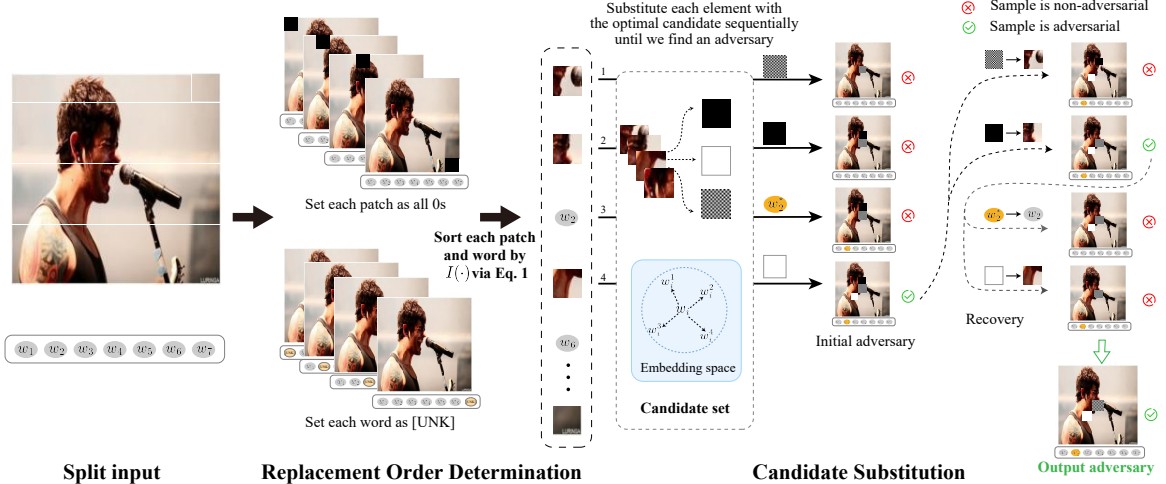

**Split input**     **Replacement Order Determination**     **Candidate Substitution**     **Output adversary**

Figure 2: The overall framework of the proposed SparseMA algorithm. SparseMA first splits the image into multiple sparse patches and the text into multiple words. Then it evaluates their importance based on the model output by setting each patch as all 0s or each word as [UNK], and determines the replacement order. According to the replacement order, SparseMA substitutes each element with a suitable candidate until we find an adversary. Finally, it randomly recovers the redundant substitution back to the original data, thereby reducing the perturbation size.

onyms based on the output logits. GA (Alzantot et al., 2018) and PSO (Zang et al., 2020) adopt evolutionary algorithms to search for a near-optimal text adversarial example.

## 2.2 Multimodal Adversarial Attack

To the best of our knowledge, there are a few works focusing on the vulnerability of multimodal models. Yu et al. (2020), Tian and Xu (2021) and Yang et al. (2021) find that samples could be misclassified by perturbing only single modality using PGD, revealing the vulnerability of multimodal models. Evtimov et al. (2020) perturb images with PGD and texts with HotFlip (Ebrahimi et al., 2018) to generate hateful posts that could fool multimodal models. MUROAN (Vishwamitra et al., 2021) utilizes the fusion features to decouple the input modalities and generate adversarial examples by removing salient data points. Co-Attack (Zhang et al., 2022) first perturbs the text and then perturbs the image according to the perturbed fusion features to conduct a collaborative attack on the pre-trained multimodal model to fool all downstream tasks.

In this work, by splitting the input image and text into multiple sparse components, SparseMA establishes a more fine-grained connection between each element rather than between the entire image and text, allowing us to generate more well-designed multimodal perturbations efficiently. To the best of our knowledge, SparseMA is the first proposed method for black-box multimodal attacks.

## 3 Methodology

In this section, we first formalize the problem of multimodal adversarial attack and then provide a detailed description of our method.

### 3.1 Multimodal Adversarial Attack

Given an input space $\mathcal{X}$ containing all input images and texts, and an output space $\mathcal{Y} = \{y_1, \ldots, y_k\}$, we have a pre-trained multimodal classifier $f : \mathcal{X} \to \mathcal{Y}$, which maps the input image $x_I$ and text $x_T = w_1 w_2 \ldots w_n$ to its ground-truth label $y \in \mathcal{Y}$. The adversary adds an imperceptible perturbation on the classified sample $x = (x_I, x_T)$ to craft an adversarial example $(x_I^*, x_T^*)$ that misleads the classifier $f$ to output a wrong prediction:

$$\underset{y_i \in \mathcal{Y}}{\arg\max} \; f(y_i | x_I^*, x_T^*) \neq y.$$

Meanwhile, we should also guarantee the imperceptibility of adversarial perturbations. Therefore, we propose evaluating the quality of generated adversarial examples using multimodal similarity. To achieve this, we input both images and texts into pre-trained BLIP-2 (Li et al., 2023) and extract the vector of the last hidden layer as the multimodal feature vector of the sample. Then, we calculate the cosine similarity between these two vectors as the multimodal similarity. This approach provides a more comprehensive evaluation of the generated adversarial examples, taking into account both visual and textual aspects of the data.

In this work, we propose a novel sparse black-box multimodal attack approach, named SparseMA, in which only the output score of the target model is needed.

### 3.2 The Proposed SparseMA Method

Given a benign sample that contains image $x_I$ and text $x_T$, SparseMA first splits image $x_I$ into multiple patches and text $x_T$ into multiple words. As illustrated in Figure 2, SparseMA consists of two main stages: *i.e.*, replacement order determination and candidate substitution. **Replacement order determination** determines the importance and replacement order of each patch and word. **Candidate substitution** sequentially substitutes each patch or word with a suitable candidate and then optimizes the redundant substitutions by random recovery to generate an optimal adversary.

#### 3.2.1 Replacement Order Determination

In order to map the image and text into a discrete space, we split the image $x_I$ into $m$ patches $\{p_1, p_2, \ldots, p_m\}$ of size $(s, s)$ and the text $x_T$ into $n$ words $\{w_1, w_2, \ldots, w_n\}$. Intuitively, more important components with greater impact on the victim model should be replaced. To evaluate the importance of each patch $p_i$ or word $w_j$, we set the patch $p_i$ to be all 0s, or set the word $w_j$ to be unknown, *i.e.*, [UNK], to obtain a modified sample $\hat{x} = (\hat{x}_I, x_T)$ or $(x_I, \hat{x}_T)$. We feed it into the target model, and calculate the importance score of each patch $p_i$ by examining the change of the model output:

$$I(p_i) = \frac{f(y|x_I, x_T) - f(y|\hat{x}_I, x_T)}{d(\hat{x}_I, x_I)}. \quad (1)$$

Similarly, we could obtain the importance score $I(w_j)$ of each word $w_j$. $I(\cdot)$ measures the impact of the unit perturbation on the victim model by taking the perturbation size $d(\cdot, \cdot)$ into account. It allows us to balance the choice of perturbation between image patches and text words to achieve maximum impact on the model with minimal disruption. In the end, we sort all patches and words in descending order based on the importance score as the replacement order.

#### 3.2.2 Candidate Substitution

To find the suitable candidate for substitution, we pre-define candidate sets for each patch and word to ensure that the replacement operation has little impact on the semantics and similarity of samples.

---

**Algorithm 1:** The SparseMA Algorithm

**Input:** Input sample that contains the image $x_I$ and the text $x_T$ with lable $y$; Target classifier $f$; Number of iterations $N$ for random recovery

**Output:** Adversarial example

/* Replacement Order Determination */
1 Split the image into $m$ patches
2 **for** *each patch in $x_I$ and each word in $x_T$* **do**
3     Compute the importance score via Eq. 1
4 Sort all patches and words based on the importance score in descending order

/* Candidate Substitution */
5 Construct the candidate set for each patch and word
6 **for** *each element in replacement order* **do**
7     Replace it on the sample $(x_I, x_T)$ with its optimal candidate via Eq. 2 to craft a new sample $(x_I, x_T)$
8     **if** $\arg\max_{y_i \in \mathcal{Y}} f(y_i|x_I, x_T) \neq y$ **then**
9        break
10 **if** $\arg\max_{y_i \in \mathcal{Y}} f(y_i|x_I, x_T) = y$ **then**
11     **return** None ;     // Attack fails
12 **for** $t = 1$ *to* $N$ **do**
13     Replace a randomly selected perturbed element on $(x_I, x_T)$ with the original data to craft a new sample $(\hat{x}_I, \hat{x}_T)$
14     **if** $\arg\max_{y_i \in \mathcal{Y}} f(y_i|\hat{x}_I, \hat{x}_T) \neq y$ **then**
15        Update $(x_I, x_T) \leftarrow (\hat{x}_I, \hat{x}_T)$
16 **return** $(x_I, x_T)$ ;     // Attack succeeds

---

For each image patch, we select all-black, all-white, and crossed black-and-white patches as its candidate set to simulate the perturbation behavior that illegal merchants are most likely to adopt in reality. For each word, we select its top $s$ nearest synonyms in the counter-fitted embedding space (Mrkšić et al., 2016) as its candidate set.

For the patch $p_i$ or word $w_j$ to be replaced on sample $x^t = (x_I^t, x_T^t)$ at the $t$-th iteration along the obtained replacement order, we greedily substitute it with the optimal candidate that has the greatest impact $\Delta P$ on the model from its candidate sets:

$$\Delta P = f(y|x_I^t, x_T^t) - f(y|x_I^{t+1}, x_T^{t+1}), \quad (2)$$

where the sample $x^{t+1} = (x_I^{t+1}, x_T^{t+1})$ is obtained by replacing the patch $p_i$ or word $w_j$ with a candi-

| Dataset | #Class | Train | Test | Avg. words | $CLIP_{ViT}$ | $CLIP_{Res}$ | ALBEF |
|---|---|---|---|---|---|---|---|
| MVSA-Single | 3 | 3,511 | 1,000 | 14 | 64.6 | 63.6 | 64.6 |
| MVSA-Multi | 3 | 17,886 | 1,000 | 14 | 76.6 | 73.4 | 72.0 |
| CrisisMMD | 7 | 17,126 | 1,000 | 118 | 95.8 | 95.3 | 89.5 |

Table 1: The statistics on the datasets and the classification accuracy (%) of victim models on the test set.

date on sample $x^t$. Then, we sequentially substitute each patch or word with its optimal candidate according to the replacement order until we find an adversarial example successfully.

Since some perturbed elements may have little influence on the initial adversarial examples, *i.e.*, there are redundant perturbations, greedy substitution would result in suboptimal adversarial examples. To further reduce the perturbation while keeping adversarial, we randomly change the perturbed patches or words back to the original data. We continue this recovery operation on the resulting sample if it is still adversarial to find an optimal adversarial example. The overall algorithm of SparseMA is summarized in Algorithm 1.

## 4 Experiments

In this section, we conduct extensive experiments to validate the effectiveness of SparseMA on three datasets and three models that are widely used.

### 4.1 Experimental Setup

**Datasets** We adopt three widely investigated datasets for vision-language multimodal classification, including MVSA-Single (Niu et al., 2016), MVSA-Multi (Niu et al., 2016) and CrisisMMD (Alam et al., 2018). MVSA-Single and MVSA-Multi are two sentiment classification datasets that contain three sentiments (positive, negative and neutral). CrisisMMD is a multimodal dataset consisting of tweets and associated images collected during seven disaster events.

**Victim Models** We consider ALBEF (Li et al., 2021b) and CLIP (Radford et al., 2021) as the victim models in this work. As CLIP is an aligned model, we construct a 2-layer multilayer perceptron for predicting the target label. Additionally, we consider two different image encoders for CLIP: ViT-B/16 (Dosovitskiy et al., 2021) and ResNet-50 (He et al., 2016), denoted as $CLIP_{ViT}$ and $CLIP_{Res}$, respectively. More details of these datasets and the classification accuracy of each model are shown in Table 1.

**Baselines** Since there are only a few multimodal attacks proposed recently, we select representative works from unimodal and multimodal black-box attacks as our baselines. For image adversarial attacks, we choose the state-of-the-art sparse black-box attack Sparse-RS (Croce et al., 2022). For text adversarial attacks, we choose the word-level black-box attacks PWWS (Ren et al., 2019) and PSO (Zang et al., 2020). For multimodal attacks, we combine Sparse-RS and PWWS to attack images and texts respectively to perform the multimodal attack, named Sp-RS&PWWS.

**Experimental Settings** The side length $s$ of image patches is set to 20, and the number of iterations $N$ for random recovery is set to four times the number of perturbed elements in the initial adversarial example. To ensure high semantic similarity between adversarial texts and benign texts for SparseMA and all text baselines, we also set the number of synonyms to 4. All evaluations are conducted on 1,000 randomly sampled texts from the corresponding test set. For a fair comparison with our SparseMA, we increase the maximum number of pixels perturbed to 3,600 and decrease the number of iterations accordingly to 2,000 for Sparse-RS.

### 4.2 Evaluation on Attack Effectiveness

To validate the effectiveness of SparseMA, we conduct experiments for vision-language classification on three victim models using three datasets. The results, including attack success rate, multimodal similarity and the number of queries, are summarized in Table 2.

We could observe that SparseMA achieves the highest attack success rate on 7 of 9 cases compared to all unimodal and multimodal baselines and is only slightly weaker than Sp-RS&PWWS in the other two cases. And we always generate adversarial examples that are more similar to the original samples, compared to Sp-RS&PWWS, indicating that we generate better adversarial examples. Meanwhile, the number of queries of SparseMA

| Model | Attack | MVSA-Multi | | | MVSA-Single | | | CrisisMMD | | |
|---|---|---|---|---|---|---|---|---|---|---|
| | | Succ. | Sim. | Query | Succ. | Sim. | Query | Succ. | Sim. | Query |
| **CLIP**ViT | Sparse-RS | 70.7 | 95.3 | 914 | 80.8 | 95.2 | 779 | 12.9 | 95.0 | 1,840 |
| | PWWS | 20.9 | 96.7 | 56 | 24.7 | 95.6 | 60 | 10.0 | 99.1 | 64 |
| | PSO | 27.6 | 97.1 | 3,427 | 29.4 | 95.4 | 3,568 | 13.3 | 98.9 | 4,443 |
| | Sp-RS&PWWS | 71.8 | 94.5 | 969 | **81.2** | 94.0 | 838 | 17.7 | 94.4 | 1,903 |
| | **SparseMA** | **82.5** | 96.6 | 388 | 80.0 | 96.9 | 366 | **27.1** | 97.1 | 557 |
| **CLIP**Res | Sparse-RS | 54.2 | 95.3 | 1,184 | 67.2 | 95.2 | 937 | 8.0 | 95.4 | 1,894 |
| | PWWS | 17.0 | 96.6 | 57 | 26.2 | 96.9 | 60 | 9.8 | 99.1 | 64 |
| | PSO | 20.8 | 96.5 | 3,702 | 30.7 | 96.0 | 3,527 | 13.3 | 99.1 | 4,446 |
| | Sp-RS&PWWS | 56.4 | 96.4 | 1,240 | 69.6 | 94.2 | 996 | 13.4 | 94.3 | 1,957 |
| | **SparseMA** | **64.5** | 96.7 | 410 | **81.1** | 96.7 | 336 | **24.2** | 97.6 | 563 |
| **ALBEF** | Sparse-RS | 22.1 | 96.7 | 1,666 | 50.4 | 97.2 | 1,258 | 30.4 | 97.2 | 1,583 |
| | PWWS | 22.3 | 97.7 | 56 | 39.2 | 95.9 | 66 | 34.4 | 99.1 | 61 |
| | PSO | 31.4 | 97.1 | 1,001 | 44.6 | 96.5 | 732 | 39.8 | 99.2 | 999 |
| | Sp-RS&PWWS | 29.1 | 95.0 | 1,721 | **61.8** | 94.9 | 1,316 | 46.5 | 96.5 | 1,643 |
| | **SparseMA** | **40.7** | 96.6 | 428 | 61.7 | 96.1 | 341 | **56.3** | 98.4 | 375 |

Table 2: Attack success rate (Succ., %) , multimodal similarity (Sim., %) and the number of queries (Query) of various attacks (image attack Sparse-RS, text attacks PWWS and PSO, and multimodal attacks Sp-RS&PWWS and SparseMA) on three models using three datasets for multimodal classification. The highest attack success rate is highlighted in **bold**. The second highest attack success rate is underlined.

is lower than that of all baselines except PWWS, being about 1/3 that of the multimodal attack Sp-RS&PWWS. Despite the lower query number and higher similarity of PWWS, our attack success rate is significantly higher than PWWS. Therefore, it cannot be concluded that PWWS implies a better effectiveness.

In conclusion, SparseMA achieves a higher attack success rate and similarity with a lower query number compared to the combined multimodal attack Sp-RS&PWWS, demonstrating the necessity of bridging the gap between different modalities for effective multimodal attack. These results validate the superiority of our proposed method. Moreover, we find that by perturbing only 2% to 3% of the top-level features, we can make the victim model output completely different predictions, thus confirming the vulnerability of the multimodal model. These discoveries suggest that the model relies heavily on these small subsets of high-level features to determine its output, but these features are highly susceptible to adversarial attacks. Therefore, improving the robustness of the model's high-level features against perturbations would be a crucial method to enhance the model's robustness.

Then, we present two instances of adversarial examples in Figure 3. It can be seen that SparseMA adds patches to the image to generate adversarial examples, which have fewer perturbations and are easier to add for attackers in the real-world. Although the perturbations are indeed visible, they do not alter the semantics of adversarial images, which are still recognizable by humans. Additionally, SparseMA perturbs fewer words than Sp-RS&PWWS. These evaluations demonstrate the high quality and practicality in the physical world of the adversarial examples generated by SparseMA.

### 4.3 Vulnerability on Different Modalities

Multimodal classifier utilizes information from various modalities to predict the classification results, which is expected to be robust to all modalities and achieves better performance. However, we find that multimodal models tend to be robust to one modality while being vulnerable to another. When attacking the robust modality, the attack performance is usually poor. In contrast, attacking the vulnerable modality would result in good attack performance. For instance, PWWS for text

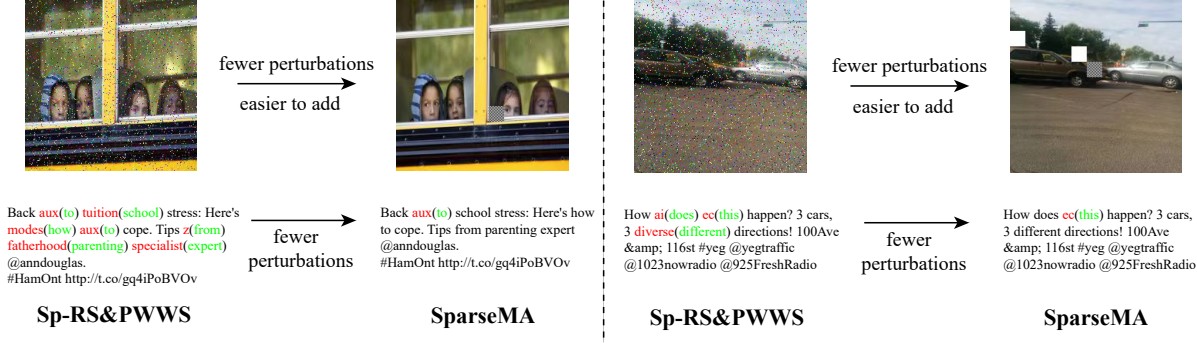

(a) The sample with positive label perturbed by various attacks is misclassified as negative.

(b) The sample with negative label perturbed by various attacks is misclassified as positive.

Figure 3: Adversarial examples generated by SparseMA and Sp-RS&PWWS on MVSA-Multi dataset. The adversarial images generated by SparseMA have fewer perturbations and are easier to add for attackers in the real-world than that of Sp-RS&PWWS. And SparseMA perturbs fewer words than Sp-RS&PWWS. We highlight the words replaced by the attacks in **Red**. The corresponding original words are highlighted in Green.

| Model | MVSA-Multi | | | MVSA-Single | | | CrisisMMD | | |
|---|---|---|---|---|---|---|---|---|---|
| | Image | Text | Both | Image | Text | Both | Image | Text | Both |
| CLIP$_{ViT}$ | 51.0 | 3.4 | 45.2 | 60.0 | 3.6 | 36.0 | 15.0 | 15.8 | 68.9 |
| CLIP$_{Res}$ | 58.0 | 1.0 | 41.0 | 56.0 | 2.5 | 41.0 | 17.0 | 17.8 | 64.9 |
| ALBEF | 20.0 | 24.0 | 56.3 | 27.0 | 28.6 | 44.4 | 12.0 | 33.9 | 54.2 |

Table 3: Percentage (%) of adversarial examples generated by SparseMA that perturb only on image or text, or both.

attack shows poor attack performance when attacking CLIP$_{ViT}$ model on MVSA-Multi dataset, as shown in Table 2. Conversely, Sparse-RS for image attack performs better. We believe this is due to the model over-relying on one modality while disregarding the other. To better evaluate the vulnerability or robustness of multimodal models to different modalities, we present the percentage of adversarial examples generated by SparseMA that perturb only on image or text, or both, in Table 3.

We can see that SparseMA prefers to perturb the more vulnerable modality. For example, SparseMA perturbs more images on the CLIP$_{ViT}$ model using the MVSA-Multi dataset, where image attacks perform well. In addition, SparseMA tends to perturb both modalities simultaneously when the multimodal model is robust to both modalities, such as on the ALBEF model using the MVSA-Multi dataset, where both image attacks and text attacks perform poorly. This demonstrates that SparseMA has discovered the vulnerability of the model and generated corresponding perturbations for the attack. It also indicates that SparseMA may be a good metric to evaluate the vulnerability or robustness of the model to different modalities.

| Attack | Clean | Brig. | Cont. | Satu. | Shar. |
|---|---|---|---|---|---|
| Sparse-RS | 70.7 | 28.0 | 30.2 | 41.5 | 32.5 |
| Sp-RS&PWWS | 71.8 | 36.5 | 39.8 | 46.7 | 39.8 |
| **SparseMA** | **82.5** | **38.9** | **40.9** | **53.4** | **45.0** |

Table 4: Attack success rate (Succ., %) of different attacks on CLIP$_{ViT}$ model using MVSA-Multi dataset when Brightness (Brig.), Contrast (Cont.), Saturation (Satu.) or Sharpness (Shar.) varies in the real-world.

## 4.4 Evaluation in the Real-World

Adversarial images captured by digital devices are usually affected by physical factors, such as brightness, saturation, contrast or sharpness, *etc*. Thus, a good physical attack should generate perturbations that are easy to be added to images, and also be resistant to various physical factors. To evaluate the effectiveness of SparseMA in the real-world, we randomly adjust the brightness, contrast, saturation, and sharpness of the adversarial images generated by various attacks on CLIP$_{ViT}$ model using MVSA-Multi dataset to $0.5$ to $1.5$ times the original value. Then we evaluate their attack success rate as shown in Table 4. The results are averaged on five runs to eliminate randomness. We could observe that brightness typically has the largest effect, and sat-

| $s$ | Succ. | Pert$_I$ | Pert$_T$ | Sim$_I$ | Sim$_T$ | Sim | Image | Text | Both | Query |
|---|---|---|---|---|---|---|---|---|---|---|
| 8 | 83.7 | 2.8 | 5.6 | 94.0 | 85.6 | 97.0 | 70.0 | 0.5 | 29.4 | 1,642 |
| 12 | **83.8** | 5.1 | 5.9 | 91.6 | 84.3 | 96.8 | 66.0 | 1.4 | 32.9 | 811 |
| 16 | 76.3 | 7.4 | 8.1 | 89.9 | 78.5 | 96.4 | 57.0 | 3.9 | 39.6 | 510 |
| 20 | 80.0 | 8.7 | 6.7 | 88.7 | 81.2 | 96.9 | 60.0 | 3.6 | 36.0 | 366 |
| 24 | 77.3 | 10.7 | 7.3 | 86.9 | 79.8 | 96.5 | 57.0 | 5.4 | 37.9 | 273 |
| 28 | 78.8 | 12.8 | 7.2 | 85.1 | 79.7 | 96.7 | 56.0 | 5.0 | 39.1 | 188 |
| 32 | 75.4 | 15.5 | 8.0 | 82.7 | 77.1 | 95.5 | 51.0 | 6.4 | 42.6 | 159 |

Table 5: Attack performance and query number (Query) of SparseMA varying different side length $s$ of the patch on CLIP$_{ViT}$ model using MVSA-Single dataset.

uration has the least effect on all attacks. Notably, SparseMA consistently achieves the highest attack success rate among all baselines, demonstrating the practicality of SparseMA in the real-world.

### 4.5 Parameter Study

To investigate the impact of the hyper-parameters in SparseMA, including the side length $s$ of patches and the number of iterations $N$ for random recovery, we conduct a series of experiments on CLIP$_{ViT}$ model using MVSA-Single dataset. We additionally evaluate the similarity to the original samples using the Structural Similarity (SSIM) (Wang et al., 2004) for images and the Universal Sequence Encoder (USE) (Cer et al., 2018) for texts, denoted as Sim$_I$ and Sim$_T$. And we adopt the perturbation rate to measure the percentage of pixels perturbed in an image or words perturbed in a text, denoted as Pert$_I$ and Pert$_T$.

**On the side length of patches** The side length restricts the minimum perturbation unit of images. In Table 5, we evaluate the attack performance of SparseMA using various side lengths from 8 to 32 with an interval of 4. SparseMA performs well when $s = 8$, achieving a high attack success rate. It also has the lowest perturbation rate and the highest similarity, which prefers to perturb images individually. However, it has a high query cost for the victim model. As we increase the side length of the image patch, the attack success rate will change unpredictably. Also, increasing the patch size results in a decrease in the number of queries but an increase in the perturbation rate. Additionally, SparseMA gradually boosts its preference for text perturbations. To balance attack performance and query cost, we select an intermediate value of $s = 20$ for our experiments.

**On the number of iterations for random recovery** To perform random recovery in the initial

adversarial example with different numbers of perturbed elements, we set the number of iterations $N$ to be an integer time of the number of perturbed elements. We evaluate the final perturbation rate and multimodal similarity in final adversarial examples using various iterations from 2 to 8 times the number of perturbed elements with an interval of 2. We observe that they achieve similar perturbation rates ($8.71\% \sim 8.78\%$ in images and $6.73\% \sim 6.92\%$ in texts) and similar multimodal similarity ($96.91\% \sim 96.94\%$). More iterations will result in a lower perturbation rate but more queries, so we choose an intermediate value, *i.e.*, $N$ is four times the number of perturbed elements.

### 5 Conclusion

We propose a vision-language black-box adversarial attack method called **Sparse Multimodal Attack** (**SparseMA**). SparseMA maps the input image and text into discrete space by splitting the image into patches and the text into words. Then it evaluates the importance of each patch or word based on its affects to the model's output. By greedily substituting important patches or words with suitable candidates, SparseMA generates adversarial examples with fewer perturbations and higher quality, which achieves a higher attack success rate than the baselines. Additionally, SparseMA can reveal the vulnerability of multimodal models to different modalities and concentrate on perturbing the more vulnerable ones. It could be served as a good metric to measure the vulnerability of multimodal models to different modalities. And it can help us better understand the multimodal model's behavior, and enhance its robustness against adversarial attacks. Experiments show that the sparse perturbations generated by SparseMA are more practical in the physical world that are easier to be added

for attackers, and perform well against real-world influences. SparseMA would be a strong baseline for future works and may inspire more researches on multimodal attacks.

## Limitations

SparseMA focuses on the two most typical modalities in the multimodal classification task, *i.e.*, continuous images and discrete text. It does not take other types of modalities, such as audio and signal, into consideration. Actually, data from these modalities can also be processed by a similar sparse strategy, and then apply our method to generate adversarial examples. We will continue to investigate the potential of SparseMA in our future work.

## Acknowledgement

This work is supported by National Natural Science Foundation (62076105, U22B2017).

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
