# OpenReview forum: "Sparse Black-Box Multimodal Attack for Vision-Language Adversary Generation"
_EMNLP/2023/Conference — EMNLP 2023 Findings_

### Official Review · Reviewer_jyg5 · 2023-07-27

**Soundness:** 2

**Excitement:**

2: Mediocre: This paper makes marginal contributions (vs non-contemporaneous work), so I would rather not see it in the conference.

**Paper Topic And Main Contributions:**

This paper investigates the robustness of vision-language multimodal classifiers, presenting a novel strategy for implementing adversarial attacks on such models. The authors' method discretizes the input image and performs attacks in the combined space of image patches and input tokens. Compared to a baseline that conducts attacks separately in image and text spaces, their approach demonstrates superior performance.

**Reasons To Accept:**

This paper proposes a novel approach for generating adversarial attacks against vision-language multimodal classifiers. Their approach performs attacks in the joint space of image patches and text tokens, and achieves superior performance against a baseline that performs attacks separately in image space and text space.

**Reasons To Reject:**

In Line 87, the authors assert that their proposed attack preserves the semantic integrity of the data. However, upon examining Figure 1, it appears that the adversarial tactics employed by illicit merchants do indeed effect semantic changes. This disparity merits further clarification from the authors.

The authors assert that their methodology represents the first-ever approach for black-box multimodal attacks. However, it's noteworthy that existing white-box attacks can potentially be transformed into black-box attacks using strategies such as Substitute Training (where a surrogate model is trained to mimic the black-box model and then attacked as a white-box model) or Zeroth-order Optimization. Thus, the distinctiveness and significance of the authors' contributions might warrant further detailed explanation.

Vision-language multimodal learning spans diverse tasks such as image captioning, visual question answering, text-to-image synthesis, image-to-text retrieval, and visual grounding. However, the authors only elucidate their focus on vision-language multimodal classification in their experiments section. The manuscript's organization could benefit from earlier clarification of this focus, as its current structure may lead to confusion for readers.

Given that vision-language multimodal classification is a niche field, the paper's contributions might be of limited impact and applicability.

The approach presented appears to draw its motivation from Figure 1. However, the connection between the problem illustrated in this figure and the investigated task of vision-language multimodal classification is not sufficiently clear.

The paper lacks sufficient implementation details for the Sp-RS&PWWS baseline. It remains unclear whether it employs an alternative optimization approach—alternating between attacks in the image space and the text space—or conducts simultaneous optimization in both spaces.

The 'multimodal similarity' metric, employed for perturbation measurement, lacks a clear definition, leaving room for ambiguity. This raises questions about the fairness and validity of the comparisons made with baseline models.

**Reproducibility:**

4: Could mostly reproduce the results, but there may be some variation because of sample variance or minor variations in their interpretation of the protocol or method.

**Reviewer Confidence:**

4: Quite sure. I tried to check the important points carefully. It's unlikely, though conceivable, that I missed something that should affect my ratings.

---

> ### Author Rebuttal · Authors · 2023-08-26
>
> We thank the reviewer for the thoughtful comments which we will address below.
>
> Question 1 The adversarial tactics employed by illicit merchants do indeed effect semantic changes
>
> Answer 1: We suspect that the reviewer may have misunderstood the application scenario of our attacks. Previous attacks have primarily focused on the imperceptibility of adversarial perturbations. In contrast, our proposed adversarial scenario is more prevalent in real-world situations, particularly in e-commerce or compliance domains. In these scenarios, attackers aim to deceive classifiers through perturbations while still allowing users to understand the non-compliant aspects of the sample, as depicted in Figure 1. We will clarify the difference in the revision.
>
> In fact, images with localized pixelation are often used on various online platforms, yet people can still effectively comprehend the content and meaning of the image. To automatically evaluate whether users can understand the information conveyed by our generated adversarial examples, we define a multimodal similarity metric similar to the commonly used text similarity tool called Universal Sentence Encoder (USE).[1]. It involves using a robust model to extract representations of samples and calculating the similarity of these representations as a measure of similarity. To extract feature vectors from multimodal inputs, we chose the BLIP2[2] model, which has demonstrated excellent performance across various downstream tasks. The experiment in Table 2 validates that our generated adversarial samples possess high quality and maintain high semantic similarity to the original samples. Therefore, although these sparse perturbations are indeed visible, they generally do not alter the semantics.
>
> As shown in Table 2, SparseMA achieves a 96% similarity between generated adversarial examples and original samples. Moreover, based on statistical analysis, we discovered that our image perturbation algorithm only affects an average of 1% to 3% of the image area. These slight perturbations generally do not alter the semantics of the samples.
>
> Question 2 The research on black-box multimodal attack
>
> Answer 2: To the best of our knowledge, there have been no reported works of transfer attacks or zeroth-order optimization black-box attacks in multimodal settings. However, direct transfer attacks of image domain often yield poor attack performance when applied to black-box models. Therefore, our work focuses on bridging the research gap in black-box attacks within the multimodal domain. By doing so, we aim to provide subsequent researchers with a valuable reference for conducting black-box adversarial attacks.
>
> Question 3: The potential applications of SparseMA on other multimodal task.
>
> Answer 3: This study serves as a mere validation of the efficacy of a multimodal attack approach, using multimodal classification as a case in point. SparseMA exhibits model-agnostic characteristics and maintains loose associations with the downstream task. Its core idea lies in establishing bridges between different modalities by means of discretization, enabling a more efficient generation of adversarial samples. This conceptual framework can equally be applied to other multimodal tasks.
>
> Question 4 Vision-language multimodal classification is a niche field
>
> Answer 4: We respectively do not agree with you. In the modern era, information is commonly presented in a multimodal manner, integrating different forms such as text and images. This can be observed in news articles, social media posts, and various other sources. Therefore, vision-language multimodal classification represents a prevalent scenario in the real world. The research on the vulnerabilities and robustness of multimodal classification models is highly valuable across various domains, including e-commerce categorization and compliance.
>
>
> Question 5: The connection between the problem illustrated in this figure and the investigated task.
>
> Answer 5: Thank you for your question. We provide a detailed explanation of the relationship between these two entities as follows. Figure 1 illustrates a scenario where illegal online merchants deceive detection models by perturbing a prohibited drug. This poses a severe security threat to numerous e-commerce companies. Therefore, our objective is to develop a sparse multimodal adversarial attack method capable of simulating these perturbation behaviors. In this paper, we employ techniques such as blacking out, whitening, and pixelation to emulate these behaviors. In practice, we achieve a 94% attack success rate on Alibaba's online detection model. We hope that SparseMA can identify more robust models or enhance the security of these applications through adversarial training. We will emphasize this connection in the revised version of our paper.
>
>
> Question 6: The implement details of Sp-RS&PWWS
>
> Answer 6: Spare-RS and PWWS are two different attacks that cannot be simultaneously applied in the same space. As a result, we apply Spare-RS in the image space and PWWS in the text space, combining the generated samples from both approaches to obtain the final adversarial examples. We will provide further clarification in the revised version.
>
>
> Question 7: About the multimodal similarity metric
>
> Answer 7: As stated in Answer 2, we utilize BLIP2 model to define a multimodal similarity metric. Although the absolute values of this multimodal similarity metric may be subject to some bias due to limitations of the model, the relative values obtained through comparison still possess significant reference value. Moreover, in most cases, SparseMA performs on par with, or even surpasses, the baselines. This observation suggests that our SparseMA approach minimally disrupts the semantic of samples and generates high-quality adversarial examples.
>
> We sincerely hope the reviewer find our response helpful, and update the scores if the concerns have been resolved. We are more than happy to further discuss any concerns that you find not fully addressed.
>
>
> [1] Universal sentence encoder

---

### Official Review · Reviewer_RAMp · 2023-08-05

**Soundness:** 3

**Excitement:**

2: Mediocre: This paper makes marginal contributions (vs non-contemporaneous work), so I would rather not see it in the conference.

**Paper Topic And Main Contributions:**

This paper proposes a vision-language black-box adversarial attack method called Sparse Multimodal Attack (SparseMA). Experiments show that the sparse perturbations generated by SparseMA are more practical in the physical world and are easier to be added for attackers.

**Questions For The Authors:**

Question A:
Can the multi-modal approach described in this paper be extended to other modalities?

Question B:
How do the authors make the adversarial perturbations imperceptible to the human eyes?

Question C:
How do the authors make sure that the semantics of the adversarial images are not changed?

**Reasons To Accept:**

1.	The performance of the proposed method is promising.

2.	This paper focuses on black-box multimodal attacks, which is an interesting and timely research topic.

**Reasons To Reject:**

1.	These sparse perturbations are visible, maybe they will change the semantics.
In Figure 2, the authors split the image into 7 × 7 blocks and then, based on the model output, evaluate their importance by setting each patch to all zeros or each word to [UNK] and determine the order of replacement. The resulting disturbance may lead to changes in the semantics of the original images. How do the authors make sure that the semantics of the adversarial images are not changed?

2.	Traditional adversarial attack methods limit the magnitude of disturbance in order to make the disturbance invisible to the human eyes. How do the authors make the adversarial perturbations imperceptible to the human eyes?

**Reproducibility:**

4: Could mostly reproduce the results, but there may be some variation because of sample variance or minor variations in their interpretation of the protocol or method.

**Reviewer Confidence:**

3: Pretty sure, but there's a chance I missed something. Although I have a good feel for this area in general, I did not carefully check the paper's details, e.g., the math, experimental design, or novelty.

---

> ### Author Rebuttal · Authors · 2023-08-26
>
> We thank the reviewer for the thoughtful comments which we will address below.
>
> Question 1: How do the authors make sure that the semantics of the adversarial images are not changed? How do the authors make the adversarial perturbations imperceptible to the human eyes?
>
> Answer 1: We suspect that the reviewer may have misunderstood the application scenario of our attacks. Previous attacks have primarily focused on the imperceptibility of adversarial perturbations. In contrast, our proposed adversarial scenario is more prevalent in real-world situations, particularly in e-commerce or compliance domains. In these scenarios, attackers aim to deceive classifiers through perturbations while still allowing users to understand the non-compliant aspects of the sample, as depicted in Figure 1. We will clarify the difference in the revision.
>
> In fact, images with localized pixelation are often used on various online platforms, yet people can still effectively comprehend the content and meaning of the image. To automatically evaluate whether users can understand the information conveyed by our generated adversarial examples, we define a multimodal similarity metric similar to the commonly used text similarity tool called Universal Sentence Encoder (USE).[1]. It involves using a robust model to extract representations of samples and calculating the similarity of these representations as a measure of similarity. To extract feature vectors from multimodal inputs, we chose the BLIP2[2] model, which has demonstrated excellent performance across various downstream tasks. The experiment in Table 2 validates that our generated adversarial samples possess high quality and maintain high semantic similarity to the original samples. Therefore, although these sparse perturbations are indeed visible, they generally do not alter the semantics.
>
> Question 2: Can the multi-modal approach described in this paper be extended to other modalities?
>
> Answer 2: Yes. As stated in limitation section, data from other modalities can also be processed by a similar sparse strategy, such as audio. Then SparseMA could efficiently perturb these samples to generate adversarial examples.
>
> Question 3: How do the authors make the adversarial perturbations imperceptible to the human eyes?
>
> Answer 3: Please see Answer 1.
>
> Question 4: How do the authors make sure that the semantics of the adversarial images are not changed?
>
> Answer 4: Please see Answer 1.
>
> We sincerely hope the reviewer find our response helpful, and update the scores if the concerns have been resolved. We are more than happy to further discuss any concerns that you find not fully addressed.
>
>
> [1] Universal sentence encoder
>
> [2] BLIP-2: Bootstrapping Language-Image Pre-training with Frozen Image Encoders and Large Language Models

---

### Official Review · Reviewer_BaP4 · 2023-08-08

**Soundness:** 4

**Excitement:**

4: Strong: This paper deepens the understanding of some phenomenon or lowers the barriers to an existing research direction.

**Paper Topic And Main Contributions:**

This paper introduces SparseMA, a black-box sparse multimodal adversarial attack approach designed to deceive multimodal deep learning models. The effectiveness of SparseMA has been evaluated in comparison with three baseline methods across three distinct datasets.

**Reasons To Accept:**

1. The research motivation is effectively presented, particularly exemplified through the insightful Figure 1.
2. The paper gives a clear algorithmic description and an illustrative overview figure of SparseMA, contributing to its comprehensibility.
3. SparseMA attains the highest attack success rate in 7 out of 9 cases, surpassing all unimodal and multimodal baseline methods.

**Reasons To Reject:**

1. The operations of Replacement Order Determination and Candidate Substitution raise concerns about significant computational overhead. Surprisingly, the paper neglects to address this issue.
2. The term "Sparse" in SparseMA remains inadequately explained within the methodology section. The rationale for expecting sparse perturbation due to such design choices is not clearly elucidated.
3. The real-world multimodal scenario depicted in Figure 1, where text is embedded in an image, lacks evaluation. The evaluation is restricted to separate images and text.

**Reproducibility:**

3: Could reproduce the results with some difficulty. The settings of parameters are underspecified or subjectively determined; the training/evaluation data are not widely available.

**Reviewer Confidence:**

4: Quite sure. I tried to check the important points carefully. It's unlikely, though conceivable, that I missed something that should affect my ratings.

**Typos Grammar Style And Presentation Improvements:**

Line 236  "Replacement order Determination" determine should be "Replacement order determination."

---

> ### Author Rebuttal · Authors · 2023-08-26
>
> We thank the reviewer for the thoughtful comments which we will address below.
>
> Question 1: The computational cost of Replacement Order Determination and Candidate Substitution
>
> Answer 1: The computational cost of adversarial attacks can be measured by the number of queries to victim the model. In our case, we can calculate that the Replacement Order Determination operation only requires 144 queries ((224/20) \* (224/20)) to determine the importance of patches, and an average of 14 queries for word importance on the MVSA-Multi dataset, totaling 158 queries. On the other hand, the Candidate Substitution operation requires a maximum of 488 queries (144 \* 3 + 14 \* 4). From Table 2, it can be observed that our query count ranges from 300 to 400. In contrast, Sp-RS&PWWS and PSO require thousands of queries. Therefore, SparseMA achieves more efficient attacks compared to Sp-RS&PWWS and PSO. We will clarify it in the revision. Thanks.
>
> Question 2: The explanation of term "Sparse" in SparseMA
>
> Answer 2: The experiments in Section 4.2 validate that Sp-RS&PWWS, which individually attacks images and text, performs worse than SparseMA, and this establishes connections between images and text. Due to their significant differences in data attributes, such as continuous images versus discrete text, it is challenging to simultaneously handle both modalities. As a result, we adopt a strategy of dividing the image into patches to convert it into discrete data, facilitating improved communication between image and text information. Meanwhile, this discretization approach allows us to handle high-resolution images more efficiently. In contrast, other sparse attacks in the image domain typically require manipulation of each pixel, resulting in a computational cost of 10^6 queries. This substantial computational cost makes them less practical and efficient.
>
> Quesiton 3: The evaluation on images where text is embedded in an image.
>
> Answer 3: In practical applications, when dealing with images like the one illustrated in Figure 1, it is common practice to employ OCR tool to extract text from the image. The extracted text is subsequently combined with the image and fed into a multimodal model for classification or attack purposes. This approach ensures that our evaluation aligns with real-world applications by leveraging both image and text information. Therefore, we can maintain consistency between our assessment and practical scenarios.
>
> Question 4: Line 236 "Replacement order Determination", determine should be "Replacement order determination."
>
> Answer 4: Thanks for your careful inspection, we'll fix it in the revised version.
>
> We sincerely hope the responses provided above address your questions. We are more than happy to further discuss any concerns that you find not fully addressed.

---

### Official Review · Reviewer_GDxW · 2023-08-10

**Soundness:** 3

**Excitement:**

2: Mediocre: This paper makes marginal contributions (vs non-contemporaneous work), so I would rather not see it in the conference.

**Paper Topic And Main Contributions:**

Authors propose a black-box attack on the semantic level against multimodal models, named Sparse Multimodal Attack (SparseMA). Motivated by malicious editing from certain online merchants, SparseMA evaluates the image patch sensitivity w.r.t. the text prompt (through the variation of model prediction) and locates the most sensitive semantic component for perturbation. The experiments show that SparseMA has higher attack success rates (ASR) compared to existing multimodal attacks.

**Questions For The Authors:**

1. What would be the major reason that the SparseMA underperforms existing baselines on the MVSA-Single dataset?

2. I stated concerns in the weakness section. Please address them to validate the work.

**Reasons To Accept:**

1. The paper is well-written with a clear description of how the framework is constructed.
2. The authors develop the framework in the black-box setting, which can potentially inspire future works against models of large parameter scales.
3. The authors explore adversarial attacks on multimodal models, which is a relatively underexplored research problem in past literature.

**Reasons To Reject:**

1. The use of importance score is questionable. The representativeness of the candidate patches depends on the variation of the model prediction score. If the model is sufficiently invariant/robust such that there are no significant prediction changes for all patches, the importance score (replacement order) will be less informative for deciding the perturbation location. In other words, Line 10 - Line 11 in Algorithm 1 can happen frequently.

2. Past adversarial works [1,2] have explored semantic attacks that manipulate attributes/appearances/textures of the object to fool the target model. A natural defense against these attacks is to perform adversarial training over the generated semantic adversarial examples. To show the validity of the work, authors should add additional discussions on the effectiveness of the proposed attack on these adversarially-trained models.

3. Dividing the input through patches to evaluate sensitivity is a brute-force approach. Authors should argue more about why the object saliency map or object segmentation is not adopted to pre-process the semantics in the image for better attack success rates.

4. There are many kinds of black-box optimization rather than the grid search adopted by the paper. Authors shall consider using more effective search algorithms (e.g., natural evolution strategy) for generating black-box attacks.

5. The paper claims the motivation concerning illegal online merchants. However, few discussions have been raised on how to defend these malicious edits using the proposed framework. It will be contributive for the paper to show the real-world applicability (e.g., perform adversarial training on this attack to get more robust models).

[1] SemanticAdv: Generating Adversarial Examples via Attribute-conditional Image Editing (ECCV 2020)

[2] Adversarial Camouflage: Hiding Physical-World Attacks with Natural Styles (CVPR 2020)

**Reproducibility:**

4: Could mostly reproduce the results, but there may be some variation because of sample variance or minor variations in their interpretation of the protocol or method.

**Reviewer Confidence:**

4: Quite sure. I tried to check the important points carefully. It's unlikely, though conceivable, that I missed something that should affect my ratings.

---

> ### Author Rebuttal · Authors · 2023-08-26
>
> We thank the reviewer for the thoughtful comments which we will address below.
>
> Question 1 The use of importance score is questionable.
>
> Answer 1: In theory, a completely robust model would exhibit minimal changes in its prediction scores when subjected to perturbations in specific patches or words. Under this scenario, the importance score (replacement order) will be less informative, as you commented. However, in practice, when we conduct adversarial attacks on existing multimodal models, we observe a high attack success rate of 80%, showing the value of importance score. This finding clearly indicates that current models are far from achieving complete robustness. The existence of such vulnerabilities underscores the significance of researching adversarial attacks.
>
> These attacks serve as a catalyst for advancing the development of models that are more robust and resistant to such adversarial attempts. By studying and understanding these attacks, we can identify weaknesses in models and work towards enhancing their robustness, ensuring the reliability and security of their predictions.
>
> Question 2 The effectiveness of the proposed attack on these adversarially-trained models.
>
> Answer 2: We appreciate your acknowledgement that attacking adversarially-trained models can highlight the effectiveness of our attack. In the real world, there are abundant samples with sparse perturbations, whereas the academic community primarily focuses on imperceptible perturbations. As a result, models adversarially trained on imperceptible perturbations may struggle to defend against adversarial samples with sparse perturbations. However, there is a lack of prior research concerning sparse attacks and adversarial training in the domain of multimodal adversarial attack now. We develop SparseMA with the intention of fostering the advancement of adversarial training against sparse perturbations. We will explore further research regarding adversarial training with SparseMA in our future work and provide a good baseline for defenses.
>
> Question 3 Is the perturbation of patch a brute-force approach?
>
> Answer 3: Perturbing objects obtained by semantic segmentation is an excellent approach to enhance the interpretability of adversarial examples. However, our investigation reveal that in black-box scenario, it can be challenging to fully understand which objects the model is focusing on. These objects may range from real-world entities like flowers, cats, and dogs to more abstract semantic objects like a smile or a logo. However current semantic segmentation models primarily concentrate on real objects, making it difficult to comprehensively cover all semantic targets. This limitation can significantly decrease the number of viable targets for perturbation and make it challenging to control the number of perturbed pixels, which can lead to a decline in the attack effectiveness.
>
> To address these challenges, we have adopted a strategy of replacing patches instead. By doing so, we can locate the regions of interest for the model without the need to specifically identify the exact objects it is focused on. Although this approach sacrifices the interpretability of adversarial samples, it greatly enhances the effectiveness of our attack.
>
> Question 4 More effective search algorithms rather than grid search
>
> Answer 4: Let me clarify that in our paper, we primarily employed a greedy algorithm rather than grid search. Greedy search we employed and other search algorithms, including genetic algorithms, possess different strengths and weaknesses. In the context of text attacks, utilizing Particle Swarm Optimization (PSO), a variant of genetic algorithms, would result in higher attack success rates but incur significant costs for accessing the model. In contrast, the greedy-based attack PWWS requires fewer queries but achieves a slightly lower attack success rate. When dealing with large multimodal models, the query cost becomes significantly high, thus motivating our selection of the greedy strategy.
>
> Question 5 The connection between the problem illustrated in this figure and the investigated task.
>
> Answer 5: Figure 1 illustrates a scenario where illegal online merchants deceive detection models by perturbing a prohibited drug. This poses a severe security threat to numerous e-commerce companies. Therefore, our objective is to develop a sparse multimodal adversarial attack method capable of simulating these perturbation behaviors. In this paper, we employ techniques such as blacking out, whitening, and pixelation to emulate these behaviors. In practice, we achieve a 94% attack success rate on Alibaba's online detection model. We hope that SparseMA can identify more robust models or enhance the security of these applications through adversarial training. We will emphasize this connection in the revised version of our paper and explore the further research regarding adversarial training with SparseMA in the future work.
>
> Question 6 The reason that SparseMA underperforms Sp-RS&PWWS on the MVSA-Single dataset?
>
> Answer 6: The parameter study in Table 5 shows that SparseMA achieves an attack success rate of 83.8% when the side length of the patch is set to 12, significantly surpassing Sp-RS&PWWS, which achieves an attack success rate of 81.2%. This may mean that the important objects in the MVSA-Single dataset are smaller, thereby requiring a smaller patch size to achieve a higher attack success rate.
>
> We sincerely hope the reviewer find our response helpful, and update the scores if the concerns have been resolved. We are more than happy to further discuss any concerns that you find not fully addressed.

---

### Meta-Review · Area_Chair_H71v · 2023-09-18

**Recommendation:** 3

**Metareview:**

This paper proposes a vision-language black-box adversarial attack method called Sparse Multimodal Attack (SparseMA). The SparseMA discretizes the input image and performs attacks in the combined space of image patches and input tokens. The effectiveness of SparseMA has been evaluated in comparison with three baseline methods across three distinct datasets. Compared to a baseline that conducts attacks separately in image and text spaces, their approach demonstrates superior performance.
In summary, all reviewers agreed that this paper focuses on black-box multimodal attacks, which is an interesting and timely research topic. The paper is well-written with a clear description of how the framework  and algorithmic are constructed. Their approach performs attacks in the joint space of image patches and text tokens, and achieves superior performance against a baseline that performs attacks separately in image space and text space. However, as three reviewers have pointed out, the effectiveness of the proposed method has not been sufficiently demonstrated, so ｍore experimental evaluations are needed to validate the contribution of this work. Specifically, the strong and significant differences with past adversarial works and the visibility of the perturbations, etc. There is a possibility that the paper will be accepted to findings.

---

### Decision · Program_Chairs · 2023-10-07

**Decision:**

Accept-Findings

**Comment:**

This paper proposes a vision-language black-box adversarial attack method called Sparse Multimodal Attack (SparseMA). The SparseMA discretizes the input image and performs attacks in the combined space of image patches and input tokens. The effectiveness of SparseMA has been evaluated in comparison with three baseline methods across three distinct datasets. Compared to a baseline that conducts attacks separately in image and text spaces, their approach demonstrates superior performance.
In summary, all reviewers agreed that this paper focuses on black-box multimodal attacks, which is an interesting and timely research topic. The paper is well-written with a clear description of how the framework  and algorithmic are constructed. Their approach performs attacks in the joint space of image patches and text tokens, and achieves superior performance against a baseline that performs attacks separately in image space and text space. However, as three reviewers have pointed out, the effectiveness of the proposed method has not been sufficiently demonstrated, so ｍore experimental evaluations are needed to validate the contribution of this work. Specifically, the strong and significant differences with past adversarial works and the visibility of the perturbations, etc. There is a possibility that the paper will be accepted to findings.